# DNBP: Differentiable Nonparametric Belief Propagation

## Abstract

We present a differentiable approach to learn the probabilistic factors used for inference by a nonparametric belief propagation algorithm. Existing nonparametric belief propagation methods rely on domain-specific features encoded in the probabilistic factors of a graphical model. In this work, we replace each crafted factor with a differentiable neural network enabling the factors to be learned using an efficient optimization routine from labeled data. By combining differentiable neural networks with an efficient belief propagation algorithm, our method learns to maintain a set of marginal posterior samples using end-to-end training. We evaluate our differentiable nonparametric belief propagation (DNBP) method on a set of articulated pose tracking tasks and compare performance with learned baselines. Results from these experiments demonstrate the effectiveness of using learned factors for tracking and suggest the practical advantage over hand-crafted approaches. The project webpage is available at: https://sites.google.com/view/diff-nbp.

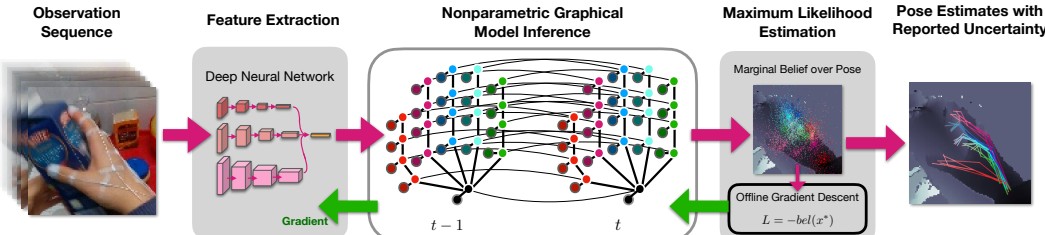

Figure 1: Architecture diagram of differentiable nonparametric belief propagation. DNBP combines domain knowledge in the form of graphical models with differentiable neural networks for tractable inference in continuous spaces. Input features from a deep neural network and the probabilistic relationships encoded in a graphical model are learned jointly in an end-to-end fashion using backpropagation. Following offline training, DNBP can be applied to unseen data without hand-tuning.

## 1 Introduction

A significant challenge for robotic applications is the ability to estimate the pose of articulated objects in high noise environments. Nonparametric belief propagation (NBP) algorithms (Sudderth et al., 2003; Isard, 2003) have proven effective for inference in visual perception tasks such as human pose tracking (Sigal et al., 2004) and articulated object tracking in robotic perception (Desingh et al., 2019; Pavlasek et al., 2020). Moreover, these algorithms are able to account for uncertainty in their estimates when environmental noise is high and show promising computational properties in practice (Desingh et al., 2019; Ortiz et al., 2021). Their adaptability to new applications, however, is limited by the need to define hand-crafted functions that describe the distinct statistical relationships in a particular dataset. Current methods that utilize NBP rely on extensive domain knowledge to parameterize these relationships. Reducing the domain knowledge required by NBP methods would enable their use in a broader range of applications.

As a form of probabilistic graphical model inference, NBP algorithms leverage domain knowledge encoded in graph-based representations, such as the Markov random field (MRF). Their capacity to

perform inference using arbitrary graphs sets them apart from other algorithms such as the recursive Bayes filter (Thrun et al., 2005) (e.g. particle filter (Godsill, 2019)) and has been shown to be important in computational perception because it allows for modeling of non-causal relationships (Sudderth et al., 2003). Data-driven approaches are an alternative for computational perception (Xiang et al., 2018; Tremblay et al., 2018). These methods generally avoid the need for extensive domain knowledge by learning from large amounts of labelled data. Data-driven approaches, however, are prone to noisy estimates and have limited capacity to represent uncertainty inherent in their estimates. In robotic applications, both of these limitations negatively impact the ability for a robot to operate effectively in unstructured environments.

In this paper, we present a differentiable nonparametric belief propagation (DNBP) method, a hybrid approach which leverages neural networks to parameterize the NBP algorithm. Through differentiable inference, DNBP leverages the explainability and robustness of probabilistic inference techniques and capitalizes on the efficiency and generalizability of data-driven approaches. Inspired by the differentiable particle filter (DPF) from Jonschkowski et al. (2018) and the pull message passing for nonparametric belief propagation (PMPNBP) algorithm (Desingh et al., 2019), we develop a differentiable nonparametric belief propagation algorithm. DNBP performs end-to-end learning of each probabilistic factor required for graphical model inference.

The effectiveness of DNBP is demonstrated on two simulated articulated tracking tasks and on a real-world hand pose tracking tasks in challenging noisy environments. An analysis of the learned probabilistic factors and resulting tracking performance is used to validate the approach. Results show that our approach can leverage the graph structure to report uncertainty about its estimates while significantly reducing the need for prior domain knowledge required by previous NBP methods. DNBP performs competitively in comparison to traditional learning-based approaches on the tracking tasks. Collectively, these results indicate that DNBP has the potential to be successfully applied to robotic perception tasks, where a notion of uncertainty in the inference is inevitable.

## 2 RELATED WORK

**Belief Propagation:** In the context of graphical models, inference refers to the process in which information about observed variables is used to derive the posterior distribution(s) of unobserved variables. Belief propagation (BP) is a message passing algorithm for inference on graphical models. BP computes exact marginal distributions on trees (Pearl, 1988), and has demonstrated empirical success on loopy graphs (Murphy et al., 1999; Sun et al., 2003; Lee et al., 2008; Lan et al., 2006). In order to apply inference techniques such as BP and LBP, the parameters of a graphical model (e.g. the probabilistic factors) must be fully specified. Maximum likelihood estimation (MLE) has been shown to be an effective approach for learning the parameters of a graphical model from data (Murphy, 2012; Koller & Friedman, 2009; Ping & Ihler, 2017). In contrast, this current study focuses on parameter learning for use with inference of continuous random variables.

**Nonparametric Belief Propagation:** For continuous spaces, such as six degrees-of-freedom object pose, exact integrals called for in BP and LBP become intractable and approximate methods for inference have been considered. Nonparametric belief propagation (NBP) methods (Isard, 2003; Sudderth et al., 2003), have been proposed which represent the inferred marginal distributions using mixtures of Gaussians and define efficient message passing approximations for inference. Isard (2003) demonstrated the effectiveness of PAMPAS using a set of synthetic visual datasets each modeled with hand-crafted factors. Sudderth et al. (2003) applied their NBP method successfully to a visual parts-based face localization task as well as a human hand tracking task (Sudderth et al., 2004). In both applications, NBP relied on factor models which were chosen based on task-level domain knowledge (e.g. valid configurations of human hands). Sigal et al. (2004) extended these NBP methods to human pose estimation and tracking using factors which were each trained separate from the inference algorithm using independent training objectives.

Ihler & McAllester (2009) described a conceptual theory of particle belief propagation, where messages being sent to inform the marginal of a particular variable could be generated using a shared proposal distribution. Following the work of Ihler and McAllester, Desingh et al. (2019) presented an efficient "pull" message passing algorithm (PMPNBP) which uses a weighted particle set to approximate messages between random variables. PMPNBP was shown to be effective on robot pose estimation tasks using hand-crafted factors. Using a similar approximation of belief propagation,

Pavlasek et al. (2020) took a step toward neural network-based potential functions by introducing a pre-trained image segmentation network to the unary factors. An important limitation of these works is they assume the probabilistic factors expressed in the graph are provided as input or rely on domain knowledge to separately model and train each function. The potential for neural networks to learn the parameters used by alternative inference techniques has been demonstrated (Do & Artières, 2010; Tompson et al., 2014). The recent work of Xiong & Ruozzi (2020) demonstrated promising inference performance on discrete classification and synthetic datasets using learned probabilistic factors with a Bethe free energy approximation. In contrast, this paper focuses on inference in high dimensional continuous state spaces using learned probabilistic factors with the particle-based inference process of NBP.

**Differentiable Bayes Filtering:** In the context of robot state estimation, many approaches have recently been proposed that incorporate neural networks with recursive inference algorithms in an end-to-end fashion. Haarnoja et al. (2016) introduced a differentiable Kalman filter, and Jonschkowski & Brock (2016) proposed a differentiable, histogram-based Bayes filter algorithm. Jonschkowski et al. (2018) and Karkus et al. (2018) both proposed differentiable particle filter algorithms for modeling continuous state spaces. Kloss et al. (2021) evaluate recent differentiable filtering techniques. Yi et al. (2021) propose an end-to-end learning method for inference over factor-graph models. In contrast to these methods, which model a single object body using variants of the Bayes filter, this work sets out to study the potential for NBP to be used as an algorithmic prior for modeling multi-part articulated objects. Recently, this line of research on differentiable state estimation algorithms has extended into the planning domain (Karkus et al., 2019; Wang et al., 2020; Anderson et al., 2019). Exploration of embedding DNBP within a differentiable planning system is left as future work.

## 3 BELIEF PROPAGATION

Consider a Markov Random Field (MRF) defined by the undirected graph $\mathcal{G} = \{\mathcal{V}, \mathcal{E}\}$, where $\mathcal{V}$ denotes a set of nodes and $\mathcal{E}$ denotes a set of edges. An example MRF model is shown in Fig. 2b. Each node in $\mathcal{V}$ represents an observed (grey) or unobserved (white) random variable, while each edge in $\mathcal{E}$ represents a pairwise relationship between two random variables in $\mathcal{V}$. The joint probability distribution for $\mathcal{G}$ is:

$$p(\mathcal{X}, \mathcal{Y}) = \frac{1}{Z} \prod_{(s,d) \in \mathcal{E}} \psi_{sd}(X_s, X_d) \prod_{d \in \mathcal{V}} \phi_d(X_d, Y_d) \tag{1}$$

where $\mathcal{X} = \{X_d \mid d \in \mathcal{V}\}$ is the set of unobserved variables and $\mathcal{Y} = \{Y_d \mid d \in \mathcal{V}\}$ is the set of corresponding observed variables. The scalar $Z$ is a normalizing constant. For each node, the function $\phi_d(\cdot)$ is the *unary potential*, describing the compatibility of $X_d$ with a corresponding observed variable $Y_d$. For each edge, the function $\psi_{sd}(\cdot)$ is the *pairwise potential*, describing the compatibility of neighboring variables $X_s$ and $X_d$. This work considers MRF models limited to pairwise clique potentials.

Given the factorization of the joint distribution defined in Eq. (1), BP provides an algorithm for inference of the marginal posterior distributions, know as the beliefs, $bel_d(X_d)$. BP defines a message passing scheme for calculation of the beliefs as follows:

$$bel_d(X_d) \propto \phi_d(X_d, Y_d) \prod_{s \in \rho(d)} m_{s \to d}(X_d) \tag{2}$$

where $\rho(s)$ denotes the set of neighboring nodes of $s$. A message from node $s$ to $d$ is defined as:

$$m_{s \to d}(X_d) = \int_{X_s} \phi_s(X_s, Y_s) \, \psi_{sd}(X_s, X_d) \times \prod_{u \in \rho(s) \setminus d} m_{u \to s}(X_s) \, dX_s \tag{3}$$

Performing inference of random variables in continuous space causes the integral in Eq. (3) to become intractable. This motivates the use of efficient algorithms that approximate the message passing scheme of Eq. (2) and Eq. (3).

### 3.1 NONPARAMETRIC BELIEF PROPAGATION

Nonparametric belief propagation (NBP) (Sudderth et al., 2003) uses Gaussian mixtures to represent the beliefs and messages for continuous random variables. Later works, including Ihler &

McAllester (2009) and Desingh et al. (2019), further improve upon the tractibility of approximate nonparametric inference by representing beliefs and messages with sets of weighted particles. These particle-based NBP methods infer an approximation of the beliefs using an iterative message passing algorithm, in which beliefs and messages are updated at each iteration $t$. In particular, Desingh et al. (2019) avoid the expensive message generation of NBP by approximating Eq. (3) with a "pull" strategy. A message, $m_{s \to d}^t$, outgoing from $s$ to $d$, is generated by first sampling $M$ independent samples from $bel_d^{t-1}(X_d)$ then reweighting and resampling from this set.

## 4 Differentiable Nonparametric Belief Propagation

We propose a differentiable nonparametric belief propagation (DNBP) method. DNBP maintains a representation of the uncertainty in the estimate by efficiently approximating the marginal posterior distributions encoded in an MRF. Our method avoids the need to define hand-crafted functions for each domain by modeling the potentials needed for the computation of the distributions with neural networks that are trained end-to-end. This hybrid generative-discriminative approach leverages the strengths of both NBP and neural networks.

DNBP uses an iterative, differentiable message passing scheme to infer the beliefs over hidden variables in an MRF. DNBP approximates the belief and messages in Eq. (2) and Eq. (3) at iteration $t$ by sets of $N$ and $M$ weighted particles respectively:

$$bel_d^t(X_d) = \left\{ \left( \mu_d^{(i)}, w_d^{(i)} \right) \right\}_{i=1}^N \tag{4}$$

$$m_{s \to d}^t = \left\{ \left( \mu_{sd}^{(i)}, w_{sd}^{(i)} \right) \right\}_{i=1}^M \tag{5}$$

DNBP relies on a "pull" message passing strategy similar to the one presented by Desingh et al. (2019). In this strategy, each iteration of the algorithm is defined in terms of a message update step and a belief update step. The message update generates a new set of message particles as a reweighted set of samples from the previous iteration's belief. Crucially, the weights associated with these updated message samples result from learned probabilistic factors as opposed to hand-crafted ones. Following a message update, the belief update combines information that is incoming to each node from the newly generated messages. Pseudocode of DNBP's message and belief update schemes is included in Appendix A.1. The following sections describe the networks used to compute the message and belief updates.

**Unary Potential Functions:** According to the factorization of the MRF joint distribution in Eq. (1), each unobserved variable $X_d$, for $d \in \mathcal{V}$, is related to a corresponding observed variable $Y_d$ by the unary potential function $\phi_d(X_d, Y_d)$. DNBP models each unary function with a feedforward neural network. The unary potential for a particle, $x_d$, given an observed image, $y_d$, is:

$$\phi_d(X_d = x_d, Y_d = y_d) = l_d \left( x_d \oplus f_d(y_d) \right) \tag{6}$$

where $f_d$ is a convolutional neural network, $l_d$ is a fully connected neural network, and the symbol $\oplus$ denotes concatenation of feature vectors. Details of network architectures are given in Appendix A.2, Table 1.

**Pairwise Potential Functions:** For any pair of hidden variables, $X_s$ and $X_d$, which are connected by an edge in $\mathcal{E}$, a pairwise potential function, $\psi_{sd}(X_s, X_d)$, represents the probabilistic relationship between the two variables. DNBP models each pairwise potential using a pair of feedforward, fully connected neural networks, $\psi_{sd}(X_s, X_d) = \{\psi_{sd}^\rho(\cdot), \psi_{sd}^{\sim}(\cdot)\}$. The pairwise *density* network, $\psi_{sd}^\rho(\cdot)$, evaluates the unnormalized potential for a pair of particles. The pairwise *sampling* network, $\psi_{sd}^{\sim}(\cdot)$, is used to form samples of node $s$ conditioned on node $d$ and vice versa. Details of network architectures are given in the Appendix A.2, Table 1. The weight computation is detailed in the pseudocode in Appendix A.1.

**Particle Diffusion:** DNBP uses a learned particle diffusion model for each hidden variable, modeled as distinct feedforward neural networks, $\tau_d^{\sim}(\cdot)$ for $d \in \mathcal{V}$. This diffusion model replaces the Gaussian diffusion models typically used by particle-based inference methods. At the outset of message generation at iteration $t$, DNBP's belief particles from iteration $t-1$ are resampled then passed through the diffusion model at the beginning of iteration $t$ to form the messages used to update the distributions at iteration $t$.

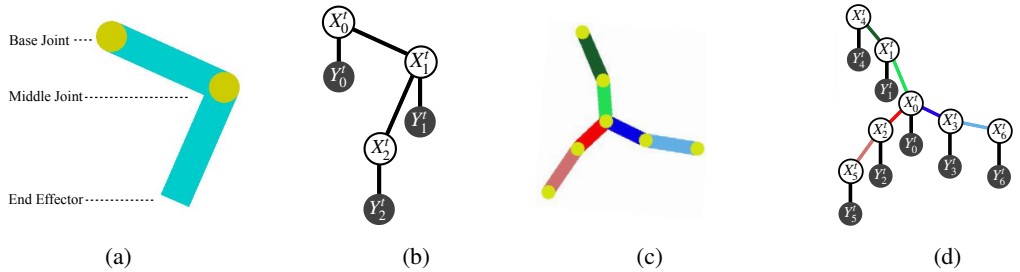

Figure 2: a) Geometry and example configuration of the double pendulum. b) Graphical model used by DNBP for the double pendulum task. c) Geometry and an example configuration of the spider structure. d) Graphical model used by DNBP for the spider task.

**Particle Resampling:** The final operation of the belief update algorithm in NBP is a weighted re-sampling of belief particles. This resampling operation is non-differentiable (Karkus et al., 2018; Jonschkowski et al., 2018). It follows that the iterative belief update algorithm is non-differentiable due to the resampling step. DNBP addresses the non-differentiability of the belief update algorithm by relocating the resampling and diffusion operations to the beginning of the message update algorithm. With this modification, the belief update returns a weighted set of particles approximating the marginal beliefs. The resulting belief density estimate is differentiable up to the beginning of the message update, when particles from the previous iteration were resampled. The resulting algorithm is differentiable through one belief update and message passing updates.

### 4.1 SUPERVISED TRAINING

DNBP's training approach is inspired by the work of Jonschkowski et al. (2018) with modifications to enable learning the potential functions distinct to DNBP. During training, DNBP uses a set of observation sequences, and a corresponding set of ground truth sequences. Using the observation sequences, DNBP estimates belief of each unobserved variable at each sequence step. Then, by maximizing estimated belief at the ground truth label of each unobserved variable, DNBP learns its network parameters by maximum likelihood estimation. Further details regarding the implementation of the training procedure are discussed in Appendix A.2.

**Objective Function:** Given a set of weighted particles representing the belief of $X_d$ produced by the inference procedure at iteration $t$, the density of the belief can be expressed as a mixture of Gaussians, with a component centered at each particle. The density of a sample $x_d$ can be computed as follows:

$$\overline{bel}_d^t(x_d) = \sum_{i=1}^{N} w_d^{(i)} \cdot \mathcal{N}(x_d; \mu_d^{(i)}, \Sigma) \tag{7}$$

DNBP defines a loss function one each hidden node $d \in \mathcal{G}$ as:

$$L_d^t = -\log(\overline{bel}_d^t(x_d^{t,*})) \tag{8}$$

where $x_d^{t,*}$ denotes the ground truth label for node $d$ at sequence step $t$. The loss for each hidden node is computed and optimized separately. At each sequence step during training, DNBP iterates through the nodes of the graph, updating each node's incoming messages and belief followed by a single optimization step of Eq. (8) using stochastic gradient descent.

## 5 RESULTS

The capability of DNBP is demonstrated on three challenging articulated tracking tasks. The first two tasks involve visually tracking the articulated joints of simulated articulated structures, as illustrated in Fig. 2. To increase the difficulty of these tasks, simulated clutter[1] in the form of static and

---

[1]In this work, clutter ratio is defined as the ratio of pixels occluded by simulated clutter to the total number of image pixels and is averaged over a full sequence of images.

dynamic geometric shapes are rendered into the image sequences. In the second task, we evaluate DNBP on its ability to track the articulated pose of human hands. In both experiments, DNBP is directly compared to learned baseline approaches that are not NBP.

## 5.1 DATASETS

**Simulated Double Pendulum:** To characterize DNBP's tracking performance under chaotic motion, the double pendulum task was chosen as an initial evaluation. The double pendulum structure consists of two revolute joints connected to two rigid-body links in series (see Fig. 2a for illustration), which are acted on by gravity. The pose of the double pendulum is modeled by the 2-dimensional position of its two revolute joints, rendered as yellow circles, and one end effector. The training set on this task consists of 1024 total sequences with 20 frames per sequence while the validation set consists of 150 total sequences with 20 frames per sequence. Both training and validation sequences are split evenly among three bins of clutter ratio: none, 0 to 0.04 and 0.04 to 0.1. Of the training and validation sequences with any amount of clutter, half contain static clutter and the other half contain dynamic clutter. The held-out test set is evenly split among clutter ratio deciles from 0 to 0.95, thus contains a shift in distribution from the training set, which was limited to clutter ratios below 0.1. Each decile contains 50 sequences with 100 frames per sequence. For test sequences with any amount of clutter, half contain static clutter and the other half contain dynamic clutter.

**Simulated Articulated Spider:** The spider task was chosen to further characterize DNBP's performance using a structure with added articulations and a larger graphical model. As depicted in Fig. 2c, the spider is comprised of three revolute-prismatic joints, three purely revolute joints, and six rigid-body links. An example of the spider is shown in Fig. 2c, in which the joints are rendered as yellow circles and the rigid-body links are rendered as coloured rectangles. Unlike the double pendulum, which contained a stationary base joint, the spider is not tethered to any position and can move freely throughout the image under simulated joint control. The training, validation and test set for this task follow the same respective distributions of clutter as were used in the double pendulum datasets. The training set consists of $2,048$ total sequences and the validation set consists of 300 sequences. The training and validation sequences are split evenly among five bins of clutter ratio: none, 0 to 0.04 and 0.04 to 0.1, 0.1 to 0.2 and 0.2 to 0.3. There are 20 frames per sequence in each of the spider datasets. Both simulated tasks use images of size $128 \times 128$ pixels. Ground truth keypoint locations are represented as continuous valued coordinates scaled to range of $[-1, +1]$.

**First-Person Hand Action Benchmark:** The FPHAB dataset (Garcia-Hernando et al., 2018) consists of RGB-D image sequences taken from the first-person perspective. Thus, the dataset captures the pose and motion of human hands as they perform typical actions. This is a challenging dataset with extreme occlusions where complete observations of all the finger joints are rare. In total, there are 1175 distinct sequences and 105459 individual image frames. Each image is labeled with the $3D$ position of 21 hand joints (illustration of joint relations shown in center column of Fig. 1). The best-performing hand pose estimation baseline proposed by Garcia-Hernando et al. (2018) is used for comparison in the current study. Just like Garcia-Hernando et al. (2018), DNBP uses only depth observations. To ensure fair comparisons with the FPHAB baseline, this study follows the 1:1 cross-subject training protocol as described in FPHAB.

## 5.2 IMPLEMENTATION DETAILS

On all three tasks, Adam (Kingma & Ba, 2015) is used for network optimization with a batch size of 6 and models are trained until convergence of the validation loss. The graphs used by DNBP are shown in Figs. 1, 2b and 2d. While DNBP uses tree-structured graphs in these experiments, the inference strategy is compatible with graphs containing cycles. DNBP is trained using 100 particles per message and tested using 200 particles per message. During training, one message update is performed at each sequence step, while two message updates are used at test time. The pairwise density, pairwise sampling and diffusion sampling processes of DNBP are defined over the relative translations between neighboring nodes. The maximum weighted particle from each marginal belief set of DNBP is used during evaluation for comparison with the ground truth.

On both simulated tasks, DNBP is compared to an LSTM recurrent neural network (Hochreiter & Schmidhuber, 1997). Both models use image inputs that are normalized channel-wise based on training set statistics. The total number of trainable parameters between LSTM and DNBP were

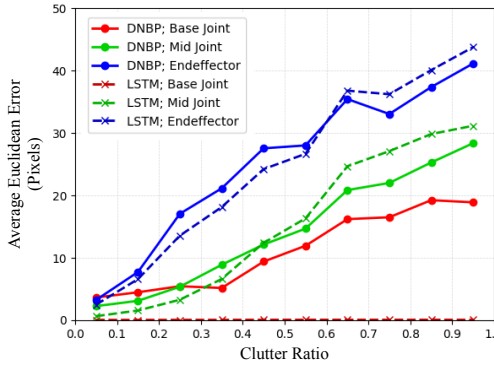 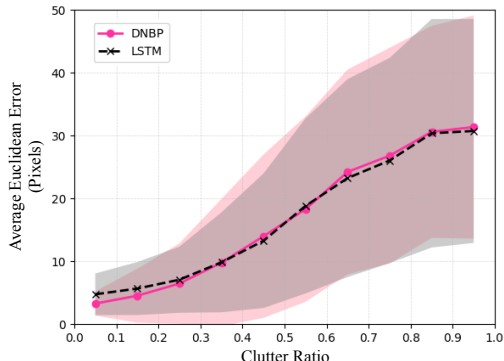

Figure 3: Average error of DNBP and LSTM predictions as a function of clutter ratio and keypoint type for double pendulum tracking.

Figure 4: Average error of DNBP and LSTM predictions as a function of clutter ratio for articulated 'spider' tracking.

chosen to be similar. For hand tracking, the preprocessing protocol of Xiong et al. (2019), is followed. Notably, preprocessing on the hand tracking task assumes ground truth bounding boxes to ensure fair comparison with the baseline method published by Garcia-Hernando et al. (2018). Similarly, the feature extractor used by DNBP in the following experiments was designed to emulate the feature extractor of compared baseline. Details of network parameters and inspection of learned relationships are included in the Appendices A.2 and A.6.

## 5.3 PERFORMANCE METRICS

As a quantitative measure of tracking error, average Euclidean error is used. On the simulated tasks, Euclidean error is averaged over all images in the test set. On the hand tracking task, Euclidean error is averaged over all joints per frame then used to calculate the percent of frames satisfying variable error thresholds as used by Garcia-Hernando et al. (2018).

Discrete entropy (Shannon, 1948) is used as a quantitative measure of uncertainty estimated by DNBP. Discrete entropy is calculated by binning samples from each marginal belief set. For qualitative analysis of the uncertainty estimated by DNBP, samples from an approximation of the joint posterior distribution (i.e. for collection of all unobserved variables) are formed using a sequential Monte Carlo sampling approach (Naesseth et al., 2014). Visualization of these samples are formed by plotting a rendered link between each pair of keypoint samples.

## 5.4 DOUBLE PENDULUM TRACKING RESULTS

As shown in Fig. 3, the keypoint tracking error of DNBP is directly compared to that of the LSTM baseline on the held-out test set for each keypoint type (base, middle and end effector) across the full range of clutter ratios. Results from this comparison show that DNBP's average keypoint tracking error is comparable to the LSTM's corresponding error for both the mid joint and end effector keypoints, independent of clutter ratio. For the base joint keypoint, which is stationary at the center position of every image, the LSTM was able to memorize the correct position. DNBP, which diffuses particles based on the message passing scheme, does not memorize the base joint position and registers a consistently larger error which increased with clutter ratio.

DNBP provides measures of uncertainty associated with its predictions, which are generated according to the algorithmic prior of belief propagation. Next tested was the hypothesis that the DNBP model would generate increased uncertainty under conditions in which an occluding object is placed into the input images such that it covers portions of the double pendulum. This test was performed by rendering an occluding block onto a test sequence as shown in Fig. 5a-c. Under optimal conditions, in which the pendulum is minimally occluded ($< 25\%$ by surface area), the model's output indicates a low level of uncertainty (see Fig. 5d,f,g.) for each keypoint and each frame. In contrast, under conditions in which the pendulum is occluded by the superimposed object, the model's output indicates relatively high levels of uncertainty precisely at frames in which the superimposed object

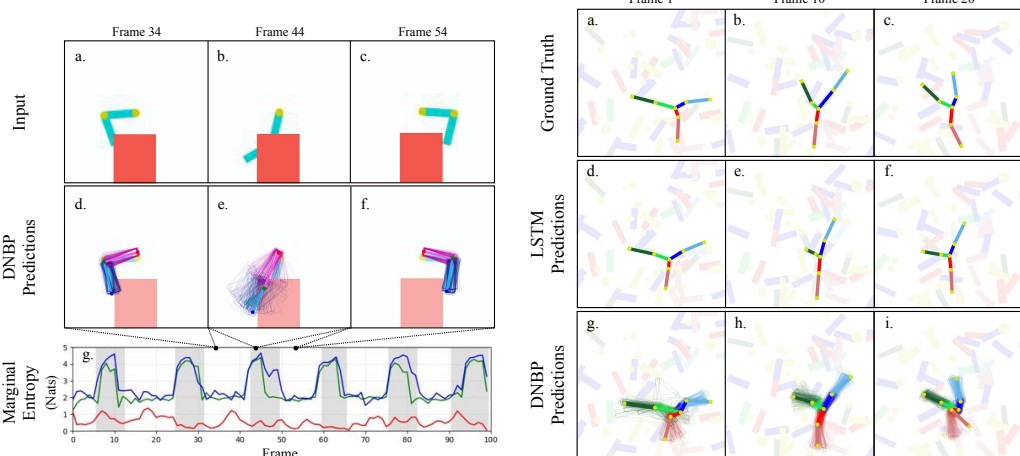

Figure 5: Tracking of double pendulum by DNBP under partial occlusion (orange block). Uncertainty associated with predictions is shown as samples from the joint distribution in pink and blue (d,e,f). (g) Marginal entropy for each keypoint across test sequence; base keypoint (red), middle keypoint (green), end-effector keypoint (blue). Sequence steps highlighted by gray correspond to images in which $> 25\%$ of the pendulum is occluded.

Figure 6: Comparison of articulated 'spider' tracking by LSTM (d,e,f) and DNBP (g,h,i) under cluttered conditions. Predicted and ground truth keypoints shown as yellow circles. Clutter shown as faded shapes for illustration to highlight predictions.

occludes a portion ($> 25\%$) of the double pendulum (see Fig. 5e,g.). These results demonstrate that the estimate of uncertainty produced by DNBP can identify predictions which are unreliable.

## 5.5 ARTICULATED SPIDER TRACKING RESULTS

After having established the performance characteristics of DNBP on the relatively straightforward double pendulum task, we next set out to determine DNBP's capability for tracking more complex structures. To this end, the 3-arm spider structure was used as a more challenging articulated pose tracking task. Each model's performance was quantitatively assessed on the held-out test set of the articulated spider tracking task using the same approach as described for the double pendulum experiment by varying clutter ratio (Fig. 4). Similar to the results of the double pendulum experiment, average error on the spider task increases as a function of clutter ratio for both the LSTM and for DNBP. For clutter ratios between $0$ and $0.25$, average error for both models remains near 6 pixels then increases consistently with clutter ratio, reaching above $30$ pixels of average error for clutter ratios above $0.85$. As in the case of the double pendulum experiment, these results demonstrate comparable performance between LSTM and DNBP on an articulated pose tracking task.

Next, a qualitative example of tracking performance under conditions of clutter is shown in Fig. 6. In Fig. 6(a-c), the ground truth pose is shown amidst distracting shapes across selected frames of a test sequence with clutter ratio of $0.25$. Pose predictions generated by LSTM are shown in Fig. 6(d-f) and by DNBP in (g-i). Qualitative assessment of the images indicates both the LSTM and DNBP place their predictions in the correct region of the image. Additionally, each model is shown to correctly predict the relative positions of the three arms. Over the sequence, both models track the motion of each keypoint, however appear to struggle with certain keypoint predictions.

## 5.6 HUMAN HAND TRACKING RESULTS

To evaluate DNBP's capability for application to real-world tasks, the algorithm's state estimation and tracking performance was evaluated on the FPHAB dataset. This is a challenging dataset with extreme occlusions where complete observations of all the finger joints are rare. Firstly, Euclidean

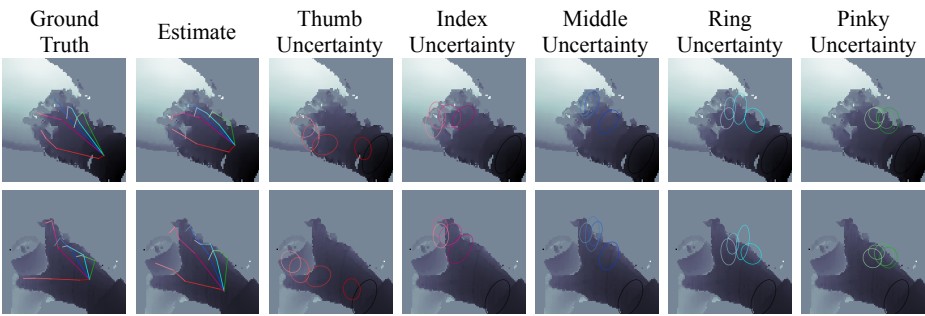

Figure 7: Output from DNBP on randomly sampled frames. See Appendix A.7 for more examples.

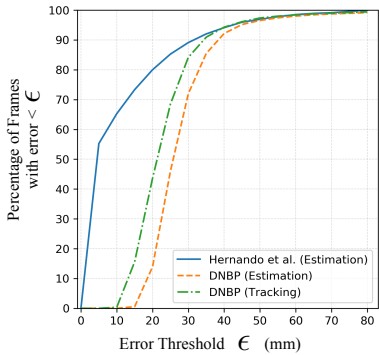

Figure 8: Quantitative comparison between DNBP and neural network baseline on hand pose tracking task of the FPHAB dataset. For each model the percent of frames with predicted pose less than a set threshold is calculated as the threshold is varied from 0mm to 80mm.

error between the estimated and ground truth pose is measured for every frame in the test set. For this first evaluation, DNBP is applied as a frame-by-frame estimator without maintaining its belief over time. The quantitative results from this experiment, are included in Fig. 8 with direct comparison to a pure neural network baseline. The results from this experiment indicate that for error thresholds below 50mm, DNBP will consistently have an accuracy of 95% and above.

Following the comparison against a state of the art baseline, it was hypothesized that DNBP's performance would improve when applied as a tracking method which maintains belief over time. To perform this test, DNBP was applied sequentially to each test sequence and evaluated under the same error metric. The result from this test, as shown in Fig. 8, demonstrates that DNBP does improve in terms of frame error when allowed to track its uncertainty over time. Qualitative examples (on frames from randomly chosen sequences) showing DNBP's tracking performance are shown in Fig. 7 and Appendix A.7. The tracking videos showing the DNBP's estimates and belief are included in the supplementary material and project webpage: https://sites.google.com/view/diff-nbp.

## 6 CONCLUSION

In this work, we proposed a novel formulation of belief propagation which is differentiable and uses a nonparametric representation of belief. It was hypothesized that combining maximum likelihood estimation with the nonparametric inference approach would enable end-to-end learning of the probabilistic factors needed for inference. Results on both qualitative and quantitative experiments demonstrate successful application of this approach and highlight the capability of DNBP to estimate useful measures of uncertainty, which are crucial for applications where incorrect estimates lead to catastrophic decisions, such as robotics. The current approach is limited by its use of non-differentiable resampling and its demand for a graph model as input. Exploration of methods to overcome these limitations, such as by incorporating a soft-resampling strategy Karkus et al. (2018), are left as future work.

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

# A APPENDIX

## A.1 ALGORITHM PSEUDOCODE

In this section, pseudo code of the DNBP message passing algorithm is given for reference. As discussed in Section 4, this algorithm is a differentiable variant of the PMPNBP algorithm Desingh et al. (2019).

---

**Algorithm 1:** Message update

**input** : Belief set $bel_d^{n-1}(X_d) = \{(w_d^{(i)}, \mu_d^{(i)})\}_{i=1}^{T}$
    Incoming messages $m_{u \to s}^{n-1}(X_s) = \{(w_{us}^{(i)}, \mu_{us}^{(i)})\}_{i=1}^{M}$ for each node $u \in \rho(s) \setminus d$

**output:** Outgoing messages, $m_{s \to d}^{n}(X_d) = \{(\mu_{sd}^{(i)}, w_{sd}^{(i)})\}_{i=1}^{M}$

1 Draw $(1 - \gamma^{n-1}) \cdot M$ independent samples from $bel_d^{n-1}(X_d)$
   $\{\mu_{sd}^{(i)} \leftarrow bel_d^{n-1}(X_d)\}_{i=1}^{(1-\gamma^{n-1}) \cdot M}$;

2 Apply particle diffusion to each sampled particle
   $\mu_{sd}^{(i)} = \mu_{sd}^{(i)} + \tau_d(\epsilon)$;

3 Draw remaining $\gamma^{n-1} \cdot M$ samples independently from uniform proposal distribution;

4 **foreach** $\{\mu_{sd}^{(i)}\}_{i=1}^{M}$ **do**

5    **for** $\ell = [1:U]$ **do**

6       Sample $\hat{X}_s^{(i)} \sim \psi_{sd}(X_s, X_d = \mu_{sd}^{(i)})$;

7       $w_{unary}^{(i)} = w_{unary}^{(i)} + \phi_s(X_s = \hat{X}_s^{(i)}, Y_s)$;

8    **end**

9    $w_{unary}^{(i)} = \frac{w_{unary}^{(i)}}{U}$;

10   **foreach** $u \in \rho(s) \setminus d$ **do**

11      $W_u^{(i)} = \sum_{j=1}^{M} w_{us}^{(j)} \times w_u^{(ij)}$ where $w_u^{(ij)} = \psi_{sd}(X_s = \mu_{us}^{(j)}, X_d = \mu_{sd}^{(i)})$;

12   **end**

13   $w_{neigh}^{(i)} = \prod_{u \in \rho(s) \setminus d} W_u^{(i)}$;

14   $w_{sd}^{(i)} = w_{unary}^{(i)} \times w_{neigh}^{(i)}$;

15 **end**

16 Associate $\{w_{sd}^{(i)}\}_{i=1}^{M}$ with $\{\mu_{sd}^{(i)}\}_{i=1}^{M}$ to form outgoing $m_{s \to d}^{n}(X_d)$;

---

**Algorithm 2:** Belief update

**input** : Incoming messages, $m_{s \to d}^{n}(X_d) = \{(w_{sd}^{(i)}, \mu_{sd}^{(i)})\}_{i=1}^{M}$, for each node $s \in \rho(d)$

**output:** Belief set $bel_d^{n}(X_d) = \{(w_d^{(i)}, \mu_d^{(i)})\}_{i=1}^{T}$

1 **foreach** $s \in \rho(d)$ **do**

2   Update message weights
     $w_{sd}^{(i)} = w_{sd}^{(i)} \times \phi_d(X_d = \mu_{sd}^{(i)}, Y_d)$ for $i \in [1:M]$;

3   Normalize message weights
     $w_{sd}^{(i)} = \frac{w_{sd}^{(i)}}{\sum_{j=1}^{M} w_{sd}^{(j)}}$ for $i \in [1:M]$;

4 **end**

5 Form belief set $bel_d^{n}(X_d) = \bigcup_{s \in \rho(d)} m_{s \to d}^{n}(X_d)$;

6 Normalize belief weights
   $w_d^{(i)} = \frac{w_d^{(i)}}{\sum_{j=1}^{T} w_d^{(j)}}$ for $i \in [1:T]$;

---

Table 1: Network parameters of learned DNBP potential functions used on both simulated articulated tracking tasks. Note $s, d \in \mathcal{V}$, and $(s, d) \in \mathcal{E}$. Unary potentials: $l_s(f_s(\cdot))$. Pairwise potentials: $\{\psi_{sd}^{\rho}, \psi_{sd}^{\sim}\}$. Particle diffusion: $\tau_s^{\sim}$.

| NETWORK | UNIT LAYERS |
|---|---|
| $f_s$ | 5 x [conv(3x3, 10, stride=2, ReLU), maxpool(2x2, 2)] |
| $l_s$ | 2 x fc(64, ReLU), fc(1, Sigmoid scaled to [0.005, 1]) |
| $\psi_{sd}^{\rho}$ | 4 x fc(32, ReLU), fc(1, Sigmoid scaled to [0.005, 1]) |
| $\psi_{sd}^{\sim}$ | 2 x fc(64, ReLU), fc(2) |
| $\tau_s^{\sim}$ | 2 x fc(64, ReLU), fc(2) |

Results in this work were generated with $U$ set to 10, while past related work (Desingh et al., 2019) used $U = 1$. It was observed that this modification improved training stability during preliminary development. Note that $\gamma$ is a hyperparameter that controls the resampling strategy and is set to 0.9 in our experiments. $\gamma$ is used only during training; during evaluation, all $M$ samples are drawn from $bel_d^{n-1}(X_d)$.

## A.2 NETWORK ARCHITECTURE & TRAINING

For both simulated articulated tracking tasks, the network architecture for each sub-network described in Section 4 is summarized in Table 1. For the hand tracking task, each network follows the same structure as those in Table 1, with two exceptions: (1) the feature extractor, $f_s(\cdot)$, used for hand tracking is based on the architecture used by the FPHAB baseline that was introduced by Ye et al. (2016). (2) each node likelihood network, $l_s(\cdot)$, has one additional feature reduction layer of [fc(64, BatchNorm,ReLU)] preceeding the layers of the corresponding network in Table 1.

The following sections describe specific implementation details used in the supervised training of DNBP. To ensure independence from spatial location, the pairwise density, pairwise sampling and diffusion sampling processes of DNBP are defined over the space of transformations between variables. Specifically, each of these networks takes as input or produces as output a translation between samples of their corresponding random variables.

**Gradient Decoupling:** The belief weight, $w_d^{t,(i)}$, of particle $i$ is proportional to the product of *component* weights, $w_{unary_d}^{t,(i)} \times w_{unary_s}^{t,(i)} \times w_{neigh_s}^{t,(i)}$, where $s$ is the neighbor of node $d$ from which particle $i$ originated (see Algorithm 2). Since each of these component weights is produced by a separate potential network (either $\phi_d$, $\psi_{sd}^{\sim}$, or $\psi_{sd}^{\rho}$ respectively), direct optimization of the belief density will lead to interdependence of the potential network gradients during training. In the context of DNBP, interdependence between different potential functions is inconsistent with the factorization given in $\mathcal{G}$. Tompson et al. (2014) describe a similar phenomenon they refer to as gradient coupling which was addressed by expressing a product of features in log-space which "decouples" the gradients.

To avoid interdependence between potential functions during training, we consider the *partial*-belief densities which are defined for each node $d \in \mathcal{V}$ as mixtures of Gaussian density functions:

$$\overline{bel}_{d,unary_d}^t(X_d) = \sum_{i=1}^{N} w_{unary_d}^{t,(i)} \cdot \mathcal{N}(X_d; \mu_d^{(i)}, \Sigma) \tag{9}$$

$$\overline{bel}_{d,unary_{\rho(d)}}^t(X_d) = \sum_{i=1}^{N} w_{unary_s}^{t,(i)} \cdot \mathcal{N}(X_d; \mu_d^{(i)}, \Sigma) \tag{10}$$

$$\overline{bel}_{d,neigh_{\rho(d)}}^t(X_d) = \sum_{i=1}^{N} w_{neigh_s}^{t,(i)} \cdot \mathcal{N}(X_d; \mu_d^{(i)}, \Sigma) \tag{11}$$

Using these definitions, direct interaction between the potential networks' gradients is avoided by maximizing the product of partial-beliefs at the ground truth of each node in log space. The product of partial-beliefs is defined:

$$\overline{bel}_d^t(X_d) = \overline{bel}_{d,unary_d}^t(X_d) \times \overline{bel}_{d,unary_{\rho(d)}}^t(X_d) \times \overline{bel}_{d,neigh_{\rho(d)}}^t(X_d) \tag{12}$$

**Unary Potentials:** During training of DNBP, only those gradients derived from the belief update of each node are used to update the corresponding node's unary potential network parameters. Any gradients derived from the outgoing messages of a particular node are manually stopped from propagating to that node's unary network. This is done to avoid confounding the objective functions of neighboring nodes, which each rely on the others' unary network during message passing. This approach can be implemented with standard deep learning frameworks by dynamically stopping the parameter update of each unary network depending on where in the algorithm its forward pass was registered.

**Pairwise Density Networks:** To speed up and stabilize the training of pairwise density potential networks, the following substitution is made during training. While calculating $w_{sd}^{(i)}$ for outgoing message $i$ from node $s$ to $d$, the summation over incoming messages from $u \in \rho(s)$ to $s$ is replaced by a single evaluation of:

$$W_u^{(i)} = \psi_{sd}(X_s = x_s^*, X_d = \mu_{sd}^{(i)})$$

(13)

where $x_s^*$ is the ground truth label of sender node $s$. This change improves inference time and reduces memory demands by removing a summation over $M$ particles while also providing more stable training feedback to the network. This substitution is removed at test time after training is complete.

**Pairwise Sampling Networks:** The pairwise sampling networks, $\psi_{s,d}^{\sim}$, take a random sample of Gaussian noise as input and generate conditional samples using the following rule:

$$\epsilon \sim \mathcal{N}(0, 1)$$

(14)

$$x_{s|d} = x_d + \psi_{s,d}^{\sim}(\epsilon)$$

(15)

where $x_{s|d}$ is the sample of variable $X_s$ conditioned on neighboring sample $x_d$ and where $\epsilon$ is a noise vector sampled from a zero-mean, unit variance multivariate Gaussian distribution with $dim(\epsilon) = 64$. Similarly, for sampling in the opposite conditioning direction (node $d$ conditioned on $s$), memory efficiency is gained by reusing the $\psi_{s,d}^{\sim}$ network but negating the sampled translation.

### A.3 DOUBLE PENDULUM CLUTTER

As summarized in Section 5.1, the double pendulum dataset was generated using a modified version of the OpenAI Brockman et al. (2016) Acrobot environment. Synthetic geometric shapes are rendered into each image of the dataset to simulate noisy, cluttered environments. All simulated clutter on the double pendulum task is generated according to the following parameters: $50\%$ of clutter is rendered visually beneath the pendulum while the remaining $50\%$ is rendered on top of the pendulum. For dynamic clutter, each geometry simulates motion using a random, constant position velocity ($\dot{x}, \dot{y}$) and orientation velocity ($\dot{\theta}$). Position velocities are sampled from $\mathcal{N}(0, 0.025)$. Orientation velocities are sampled from $\mathcal{N}(0, 0.05)$. Clutter is simulated as either rectangles with $80\%$ probability or circles with $20\%$ probability. Clutter rectangles are sized randomly with length of $\max(0, l \sim \mathcal{N}(0.2, 0.05))$ and height of $\max(0, h \sim \mathcal{N}(0.8, 0.2))$. Color of clutter rectangles is randomly chosen with RGB of $(0, 204, 204)$ or $(245, 87, 77)$. Clutter circles are sized randomly with radius of $\max(0, r \sim \mathcal{N}(0.1, 0.1))$ and colored randomly with RGB of $(204, 204, 0)$ or $(96, 217, 63)$. Size and color distributions were chosen to ensure clutter visually resembles the double pendulum parts. The position of each clutter geometry was randomly initialized within 1.5x the extent of the image boundary.

The training and validation datasets were distributed evenly among clutter ratios of $[0, 0 - 0.04,$ and $0.04 - 0.1]$. For the training/validation sequences that included clutter, the number of clutters rendered beneath and on top of the double pendulum was individually randomly sampled from independent Binomial distributions using $n = 15$, $p = 0.3$. To generate the test set, which was uniformly distributed among clutter ratios as described in Section 5.1, rejection sampling was used with variable numbers of rendered geometries.

### A.4 ARTICULATED SPIDER MODEL

Data for the articulated spider tracking task of Section 5.5 was simulated using the Pillow Clark (2015) image processing library. Three revolute-prismatic joints are all centrally located and treated

as the root of the spider's kinematic tree. The remaining three revolute joints are attached to pairs of links, forming three distinct 'arms' of the spider. Each joint is rendered as a yellow circle while the six rigid-body links are rendered as distinct red, green or blue rectangles respectively. Size parameters that follow are with respect to rendered image size of 500x500px. The three inner revolute-prismatic joints include rotational constraints limiting each to a non-overlapping $120°$ range of articulation as well as prismatic constraints limiting the extension to within $[20, 80]$ pixels of translation. The three purely revolute joints are constrained to rotations between $\pm 35°$ with respect to their local origins. Each rigid-body link has width of 20px and height of 80px pixels while each joint has radius of 10px.

For every simulated sequence, the spider is initialized with uniformly random root position within the central 180x180px window and uniformly random root orientation from $[0, 2\pi]$. Furthermore, each joint state is initialized uniformly at random within its particular articulation constraints. The spider is simulated with dynamics using randomized, constant root, and joint velocities with respect to a time step ($dt$) of $0.01$. The root's position velocities ($\dot{x}$, $\dot{y}$) are each sampled from an equally weighted 2-component Gaussian mixture with means ($+24$, $-24$) and standard deviations ($15$, $15$). Whereas, the root's orientation velocity ($\dot{\theta}$) is sampled from an equally weighted 2-component Gaussian mixture with means ($+0.3$, $-0.3$) and standard deviations ($0.1$, $0.1$). Each rotational joint's velocity is sampled from an equally weighted 2-component Gaussian mixture with means ($+0.3$, $-0.3$) and standard deviations ($0.1$, $0.1$). Similarly, each prismatic joint's velocity is sampled from an equally weighted 2-component Gaussian mixture with means ($+500$, $-500$) and standard deviations ($60$, $60$). Note that if any joint reaches an articulation limit during simulation, the direction of its velocity is reversed.

### A.5 ARTICULATED SPIDER CLUTTER

Clutter generation for the articulated spider tracking task follows a similar generation process as was used for the double pendulum task. Clutter parameters that follow are with respect to rendered image size of 500x500px and time step ($dt$) of $0.01$. $50\%$ of clutter is rendered beneath and $50\%$ is rendered on top of the spider. For dynamic clutter, each geometry simulates motion using a random, constant position velocity ($\dot{x}$, $\dot{y}$) and orientation velocity ($\dot{\theta}$). Position velocities are sampled from $\mathcal{N}(0, 3)$ while orientation velocities are sampled from $\mathcal{N}(0, 0.05)$. Clutter is simulated as either a rectangle with $70\%$ probability or a circle with $30\%$ probability. Clutter rectangles are sized randomly with length of $\max(0, l \sim \mathcal{N}(20, 3))$ and height of $\max(0, h \sim \mathcal{N}(80, 5))$. The color of clutter rectangles is chosen uniformly at random from the same colors as were used for the spider arms. Clutter circles are sized randomly with a radius of $\max(0, r \sim \mathcal{N}(10, 3))$ and colored yellow to match the color of the spider's joints. The position of each clutter geometry was randomly initialized within the image boundary.

For the training/validation sequences that included clutter, the number of clutter shapes rendered beneath and on top of the double pendulum was each randomly sampled from independent Binomial distributions using $n = 10$, $p = 0.5$. The test set was generated with uniformly distributed clutter ratios, as described in Section 5.1, using rejection sampling with variable numbers of rendered geometries.

### A.6 LEARNED PAIRWISE INSPECTION

As further validation of DNBP, the learned pairwise potentials are inspected in Fig. 9. The normalized histogram of pairwise translations computed from the training set for $X_1 - X_0$ (top) and $X_2 - X_1$ (bottom) are shown in the left column of Fig. 9. The middle column shows the normalized histogram of samples from learned pairwise sampler networks, $\psi_{0,1}^{\sim}(\cdot)$ and $\psi_{1,2}^{\sim}(\cdot)$. Finally, the right column shows output from the learned pairwise density networks, $\psi_{0,1}^{\rho}(\cdot)$ and $\psi_{1,2}^{\rho}(\cdot)$, generated with 100x100 uniform samples across pairwise translation space. The qualitative similarity between each learned potential model and the corresponding true distribution of pairwise translations indicates that DNBP is successful in learning to model each pairwise potential factor. The circular pairwise relationships are explained by the fact that each pair of double pendulum keypoints is related by a revolute joint. The effect of simulated gravity in the double pendulum experiment can be observed by the bias of each pairwise potential in favor of the lower half of each plot as indicated by increased likelihood.

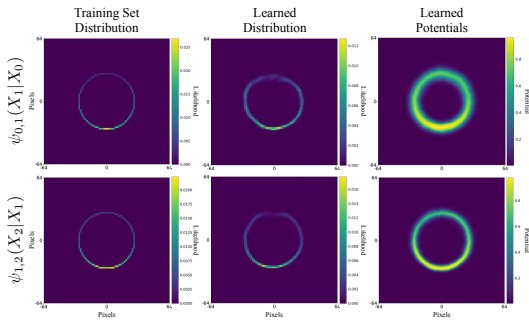

Figure 9: Inspection of learned pairwise potentials from double pendulum tracking.

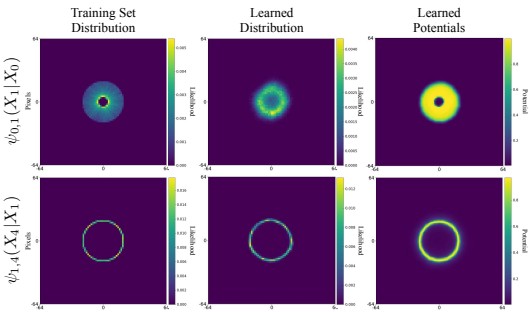

Figure 10: Inspection of DNBP's learned pairwise potentials from spider tracking. Only two of the six are shown to avoid redundancy, remaining four show very similar output.

The pairwise potential functions learned by DNBP in the spider tracking task are visualized as was done in the double pendulum task. Fig. 10 shows qualitative output from two of the six models. Only two are shown to avoid redundancy; chosen results are representative of remaining four potential functions. The left column of Fig. 10 shows the normalized histogram of pairwise translations as computed from the training set for $X_1 - X_0$ (top) and $X_4 - X_1$ (bottom). The middle column of Fig. 10 shows the normalized histogram of samples from learned pairwise sampler networks, $\widetilde{\psi}_{0,1}(\cdot)$ and $\widetilde{\psi}_{1,4}(\cdot)$. Finally in the right column of Fig. 10, uniformly sampled output (100x100 samples across pixel space) of the learned pairwise density networks is shown. Once again, the visual similarity between each learned potential function and the corresponding true distribution of pairwise translations is an indicator that DNBP is successful in learning to model each pairwise factor. Observe that the learned potential functions for $\psi_{1,4}(\cdot)$, which correspond to a revolute articulation, show no bias in favor of the downward configuration. This result is notably different from the potential functions learned on the double pendulum task and can be explained by the absence of gravity in the spider simulation. Similarly, the learned models for $\psi_{0,1}(\cdot)$ on the spider task exhibit a torus shape due to the effect of prismatic motion associated with the corresponding joint's articulation type and constraint.

## A.7 HAND TRACKING RESULTS

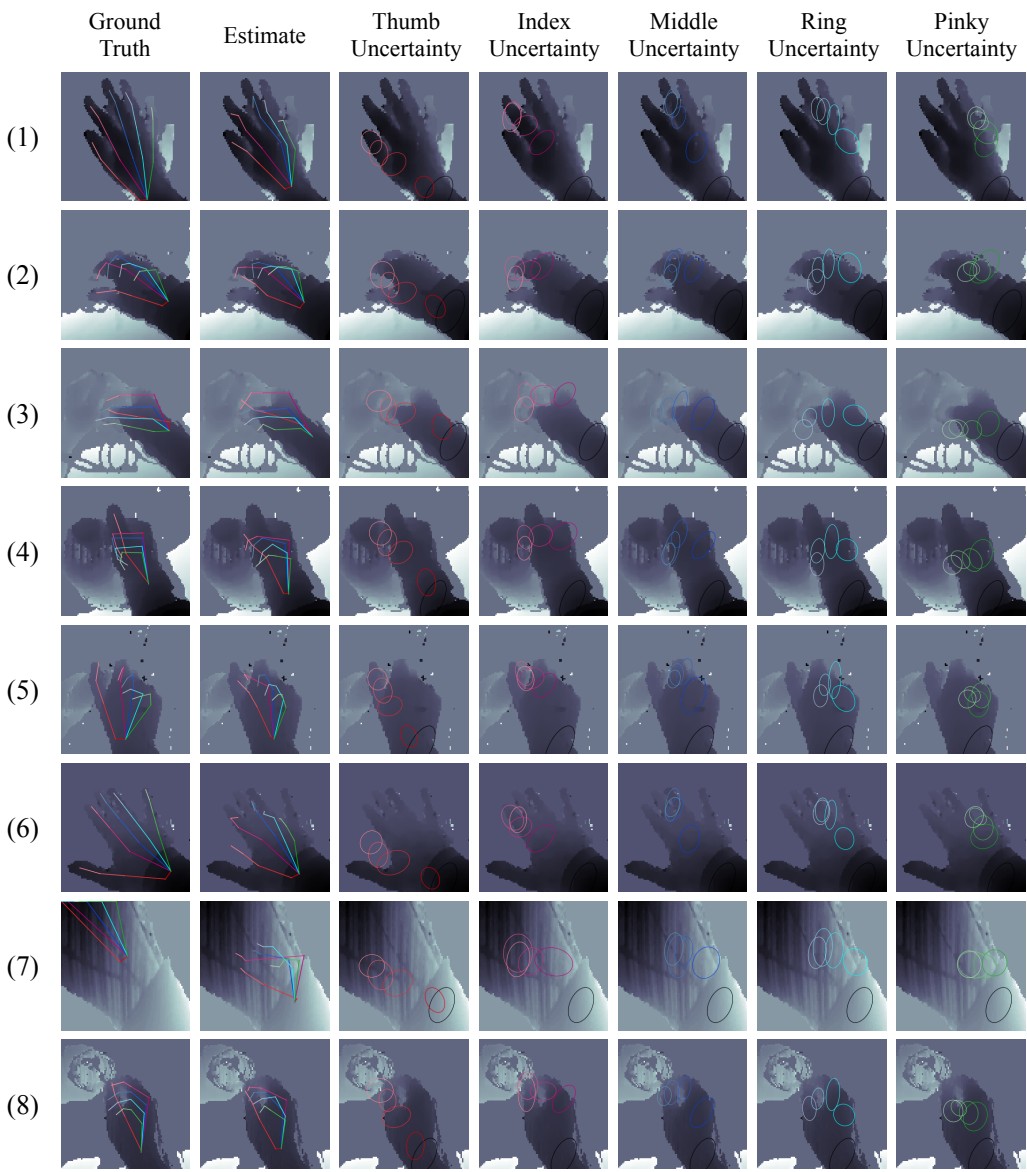

Figure 11: Output from DNBP on randomly sampled frames of the hand pose tracking experiment. Visualized model uncertainty is calculated from the marginal belief estimates of DNBP as 1 standard deviation in the $x$ and $y$ dimensions respectively as calculated by estimated covariance of belief particles. Uncertainty in depth dimension is not visualized.

