# OpenReview forum: "DNBP: Differentiable Nonparametric Belief Propagation"
_ICLR.cc/2022/Conference — ICLR 2022 Submitted_

### Official Review · Reviewer_sUCc · 2021-10-29

**Correctness:** 4
**Technical Novelty And Significance:** 2
**Empirical Novelty And Significance:** 2
**Recommendation:** 3
**Confidence:** 4

**Main Review:**

## Strengths

1. The idea of replacing the handcrafted potential functions with feed-forward neural networks is interesting.

2. The paper is well written and easy to read.

3. The implementation of the gradients is not straightforward, i.e it does not simply consist of building the computational graph and back-propagating through it.

## Weaknesses

1. The main weakness to me, which is also pointed out in the conclusion, is the fact that the graph structure needs to be provided (i.e. it is handcrafted). Moreover, if I am not mistaken, the paper only considers trees (i.e. graphs without loops). These two limitations are (likely) the reason why DNBP is not able to outperform the considered baseline in each application (it is even significantly outperformed by the method Hernando et al. on the hand pose tracking task).

2. In section 4 "Pairwise potential functions", it looks like the pairwise sampling network $\psi_{sd}^{\sim}$ output is deterministic. Shouldn't it be stochastic (since it is a "sampling" network)? Please provide more details about this network.

3. The main benefit of DNBP over classical neural networks, is its ability to associate a measure of uncertainty to its predictions but the three applications that are considered do not accentuate this benefit. I suggest to evaluate DNBP on an application where its ability to associate a measure of uncertainty is very important/useful.

**Summary Of The Paper:**

The method (DNBP) proposed in the paper considers a non-parametric belief propagation method where the unary potential functions, the pairwise potential functions and the particle diffusion function are modeled as feed-forward neural networks. It allows them to learn the parameters of these networks using labeled data which is the main contribution of the paper.

DNBP is evaluated on three tasks (simulated double pendulum, simulated articulated spider, first-person hand action). In each application, DNBP is not able to outperform the considered baseline but it is able to provide measures of uncertainty associated with its predictions.


**Summary Of The Review:**

Currently, the weaknesses outweigh the strengths. Especially the fact that the handcrafted graph structure is probably the reason why DNBP is not able to outperform the considered baselines.

---

> ### Author Response · Authors · 2021-11-23
> **Response to Reviewer sUCc**
>
> We thank Reviewer sUCc for their constructive and thoughtful feedback and address each point below.
>
> > The main weakness to me, which is also pointed out in the conclusion, is the fact that the graph structure needs to be provided (i.e. it is handcrafted). Moreover, if I am not mistaken, the paper only considers trees (i.e. graphs without loops). These two limitations are (likely) the reason why DNBP is not able to outperform the considered baseline in each application (it is even significantly outperformed by the method Hernando et al. on the hand pose tracking task).
>
> We acknowledge that graph structure learning is an interesting and important topic for MRF inference. It is possible that DNBP’s performance could be improved by extending the model to enable graph structure learning and this possibility could be explored in future work. Nevertheless, this limitation does not diminish the value of this report showing for the first time DNBP’s capacity to learn the potential functions given a graph structure and opening up this new line of research.
>
> Applying DNBP to loopy versions of the graphs could potentially improve the model’s performance and will be considered in future iterations of this work.
>
> > In section 4 "Pairwise potential functions", it looks like the pairwise sampling network $\psi_{sd}^{∼}$
>  output is deterministic. Shouldn't it be stochastic (since it is a "sampling" network)? Please provide more details about this network.
>
> We have added a discussion of these networks to the Appendix A.2. Crucially, the pairwise sampling networks, $\psi_{sd}^{∼}(\cdot)$, are themselves deterministic functions used to approximate a sample of the variable $X_s$ conditioned on a sample of $X_d$. As input, these networks process a random sample of zero-mean, unit variance Gaussian noise, $\epsilon\sim\mathcal{N}(0,1)$, to produce a sampled translation as output. Finally, the sampled translation is applied to the conditioning sample according to the following formula: $x_{s|d}=x_d + \psi_{s,d}^{∼}(\epsilon)$.
>
> > The main benefit of DNBP over classical neural networks, is its ability to associate a measure of uncertainty to its predictions but the three applications that are considered do not accentuate this benefit. I suggest to evaluate DNBP on an application where its ability to associate a measure of uncertainty is very important/useful.
>
> We have revised the paper’s conclusion to discuss the benefit of DNBP’s uncertainty estimates in relation to specific application domains. For example in the case of a robot intending to interact with a human hand, having access to a measure of uncertainty of the perceived human hand pose is crucial for the robot to avoid taking dangerous misactions.

---

> > ### Comment · Reviewer_sUCc · 2021-11-29
> > **Response**
> >
> > Thank you for answering my questions.

---

### Official Review · Reviewer_Yvfb · 2021-11-02

**Correctness:** 2
**Technical Novelty And Significance:** 3
**Empirical Novelty And Significance:** 2
**Recommendation:** 3
**Confidence:** 3

**Main Review:**

Strengths:
For readers who aren't sufficiently knowledgeable on BP techniques, the paper does a good job of going over the basics of BP.
The paper is clearly written, easy to understand.

Weaknesses:
The main issue I have with the paper is that I find the results of the paper weak:
1) The chosen baseline is LSTM, and it is trained with an extremely small batch size (6). This doesn't provide enough confidence that the DNBP is any better than a simple neural approach. Quantitative differences are very small, as are the toy datasets.
2) LSTM is not the natural choice for graphical models. Graph Neural Networks (GNN) would be far more appropriate. In fact, GNNs are shown to beat BP techniques significantly in this uncited paper: Yoon et al. "Inference in Probabilistic Graphical Models by Graph Neural Networks".
3) Why is DNBP not compared against NBP or some other BP method? The paper mentioned above does compare against BP for instance. The paper doesn't give a good reason to choose this method over others.
4) Qualitatively, hand tracking results are significantly worse than any other recent hand tracking method I know, so it is hard to judge how well the method works. The jitter is extremely high, which would be the opposite of what I'd expect from a method that takes uncertainties into account.
5) I'd expect the uncertainty estimates to be elongated along the bones for the hand tracking results. For some reason DBNP seems to somehow claim that keypoints can be significantly outside of the hand area. Especially for a depth based method, where background pixels are very far away, this doesn't make any sense. Likewise for the toy dataset results with occlusion, DNBP doesn't seem to do a better job than the simplistic LSTM.
6) Other Neural Net based methods that can measure uncertainty are not mentioned. For instance, Kumar et al.'s "LUVLi Face Alignment: Estimating Landmarks' Location, Uncertainty, and Visibility Likelihood" paper uses Gaussian and Laplacian log-likelhood losses, and their uncertainty estimates align much better with the face features, e.g. elongated along the jawline. Here we don't see the same property. Also that paper has a good methodology to check how informative the uncertainties are. I don't see a similar analysis in this paper.






**Summary Of The Paper:**

This paper enables end-to-end learning of the factors of a graphical model for nonparametric belief propagation (NBP) methods by using neural networks. It calls this method "Differentiable nonparametric belief propagation" (DNBP).

The aim is to replace domain-specific hand crafted factors with learned factors, by replacing each factor with a neural network. Compared to vanilla neural net based solutions, DNBP also reports uncertainty.

The method is evaluated on a couple of toy examples of articulated pose tracking, as well as using hand pose estimation on the FPHAB dataset.

The method is compared against learned, neural network based baselines.


**Summary Of The Review:**

While I'm not an expert on BP techniques and haven't verified the math (hence the low confidence), I've found the paper to be weak on experimental results. The LSTM baselines are very weak, and some important citations are missing. Uncertainty visualizations for hands do not seem sensible, and the uncertainty estimates are not analyzed in depth. There are no comparisons with other BP methods, as well as other uncertainty prediction techniques. Even if the approach seems sound, the results don't justify adopting this challenging-to-implement method over any other regular method.

---

> ### Author Response · Authors · 2021-11-23
> **Response to Reviewer Yvfb**
>
> We thank Reviewer Yvfb for their thorough review and address each point below.
>
> > The chosen baseline is LSTM, and it is trained with an extremely small batch size (6). This doesn't provide enough confidence that the DNBP is any better than a simple neural approach. Quantitative differences are very small, as are the toy datasets.
>
> We acknowledge that DNBP did not outperform the neural network baselines in terms of error. The goal of this work was not to demonstrate DNBP achieving superior performance over neural network baselines, but rather to identify and validate a new learning strategy to replace the hand-crafted factors required by existing nonparametric belief propagation algorithms.
>
> > LSTM is not the natural choice for graphical models. Graph Neural Networks (GNN) would be far more appropriate. In fact, GNNs are shown to beat BP techniques significantly in this uncited paper: Yoon et al. "Inference in Probabilistic Graphical Models by Graph Neural Networks".
>
> We appreciate this suggestion to compare DNBP with graph neural networks. While the GNN model proposed by Yoon et al. (2019) demonstrated superior performance when compared to a BP model, the evaluation was performed only on binary MRFs. In contrast, our work focuses on the more challenging task of inference in continuous state spaces, where a number of studies have demonstrated BP models achieving impressive performance. We agree that an interesting direction for future work would compare DNBP to GNN models on continuous state space inference tasks.
>
> > Why is DNBP not compared against NBP or some other BP method? The paper mentioned above does compare against BP for instance. The paper doesn't give a good reason to choose this method over others.
>
> The experiments in this paper were chosen such that hand-crafted factors required for traditional nonparametric belief propagation methods could not be easily defined. The BP approach compared against in Yoon et al. (2019) is applied only to binary MRF state spaces and therefore is not directly comparable to DNBP on continuous state space inference.
>
> > Qualitatively, hand tracking results are significantly worse than any other recent hand tracking method I know, so it is hard to judge how well the method works. The jitter is extremely high, which would be the opposite of what I'd expect from a method that takes uncertainties into account.
>
> The baseline hand tracking model (Garcia-Hernando et al., 2018) was used in our comparisons to validate DNBP’s hand tracking performance. This baseline was chosen because to the best of our knowledge, it is the top performing model on the selected hand tracking dataset. We agree that an interesting direction for future work could potentially improve DNBP’s performance by smoothing the particle-based estimates produced by DNBP.
>
> > I'd expect the uncertainty estimates to be elongated along the bones for the hand tracking results. For some reason DBNP seems to somehow claim that keypoints can be significantly outside of the hand area. Especially for a depth based method, where background pixels are very far away, this doesn't make any sense. Likewise for the toy dataset results with occlusion, DNBP doesn't seem to do a better job than the simplistic LSTM.
>
> We thank reviewer Yvfb for this observation. In reference to the qualitative hand tracking examples (Fig. 7 and Fig. 11), DNBP’s uncertainty is visualized as 1 standard deviation in the estimated covariance matrix. These results suggest that DNBP does maintain non-zero probability mass outside of the hand region, which may be a useful characteristic for datasets with a high degree of clutter and occlusion, such as the FPHAB hand tracking dataset. An important direction for future work could be to establish, quantitatively, how well calibrated the estimates of DNBP are.
>
> > Other Neural Net based methods that can measure uncertainty are not mentioned. For instance, Kumar et al.'s "LUVLi Face Alignment: Estimating Landmarks' Location, Uncertainty, and Visibility Likelihood" paper uses Gaussian and Laplacian log-likelhood losses, and their uncertainty estimates align much better with the face features, e.g. elongated along the jawline. Here we don't see the same property. Also that paper has a good methodology to check how informative the uncertainties are. I don't see a similar analysis in this paper.
>
> We appreciate this suggestion to consider alternative neural network models that can estimate uncertainty. Since the LUVLi model proposed by Kumar et al. (2020) is tailored to the problem of 2D face alignment and uses a parametric model of uncertainty, it is not directly comparable to DNBP on the experiments considered. However, we agree that additional analyses to understand how informative DNBPs uncertainty estimates are is an important direction for future work.

---

### Official Review · Reviewer_Rbn6 · 2021-11-02

**Correctness:** 3
**Technical Novelty And Significance:** 2
**Empirical Novelty And Significance:** 2
**Recommendation:** 3
**Confidence:** 4

**Main Review:**

**Strengths:**
 The paper aims at a differentiable approach for supervised learning and prediction in MRFs with infinite state spaces.

 **Weaknesses:**
 It is well known that loopy BP inherently fails in estimating pairwise marginals for MRFs on graphs with cycles. The examples analysed in experiments are MRF on trees. The authors do not clearly state that in such cases there is no need for loopy BP.

 The proposed BP approach employs a series of approximations which are not concisely explained. Some of them are given in the supplements only. E.g.:
  - what is meant by pairwise sampling network? (Sec. 4 and line 6 of the Algorithm in the supplement)
  - The pull strategy approximation is adopted from Desingh et al. (2019). The quality of this approximation is not evaluated.
  - Further empirical approximations of the BP approach (which is itself an approximation) are explained in the supplement even though they are part of the proposed approach. Again, their approximation quality is not evaluated.

Overall, I can imagine that such "approximations of approximations" can be useful if the main goal is to solve a particular application and they are unavoidable. However, this is not adequate if the goal is to develop a conceptual approach.

The authors mention the work Xiong & Ruozzi, (2020), where the task of computing unary and pairwise marginals is solved approximately by variational inference and then combined with learning of MRF potentials given by neural networks. This approach has the potential to learn MRFs on graphs with cycles. I would have expected at least an experimental comparison with this method.

Furthermore, it is well known that supervised learning of MRFs with finite state spaces can be approached by using the Pseudo-likelihood estimator instead of the ML estimator. This obviates the need of computing unary and pairwise marginals during learning. I would expect that it is much easier to adapt the PL estimator to the case of infinite state spaces.

The artificial examples studied in the experiments and seemingly also the human hand tracking use MRFs on trees. This would obviate the need  for loopy BP, because in this case a standard two-pass BP can be used instead.

It seems that the proposed approach is worse (w.r.t. estimation) than the neural network baseline (Fig. 8)? This holds both, for the frame by frame variant and the tracking variant of the proposed approach? It remains unclear to me how the latter tracking variant works precisely.

**Summary Of The Paper:**

Supervised learning of Markov Random Fields (MRF) by MLE requires to compute pairwise and unary marginals of the current model estimate at each iteration of the likelihood maximisation. This task is not tractable except for MRFs on trees with finite hidden state spaces. The authors consider MRFs with infinite state spaces, propose to model the pairwise and unary potentials by neural nets and aim at developing an approximated belief propagation (BP) approach for learning these networks. BP is known to be exact on trees but is not tractable for infinite state spaces. The authors apply their method on a challenging task of hand pose estimation in  RGB-D image sequences taken from the first-person perspective.

**Summary Of The Review:**

The proposed approach for learning MRF parameters by combining neural networks with an approximated belief propagation method is not concisely explained and involves (in my view) too many ad-hoc approximations. The experimental results are not convincing enough to compensate the conceptual weaknesses.

---

> ### Author Response · Authors · 2021-11-23
> **Response to Reviewer Rbn6 (1/2)**
>
> We thank Reviewer Rbn6 for their constructive review and address each point below.
>
> > It is well known that loopy BP inherently fails in estimating pairwise marginals for MRFs on graphs with cycles. The examples analysed in experiments are MRF on trees. The authors do not clearly state that in such cases there is no need for loopy BP.
>
> We acknowledge that loopy BP is an approximate inference strategy when applied to graphs with cycles. The graphs used in our experiments were chosen based on the known kinematic structure of each articulated object. We have revised Section 5.2 to discuss our choice of graphs.
>
> > what is meant by pairwise sampling network? (Sec. 4 and line 6 of the Algorithm in the supplement)
>
> We have added a discussion of these networks to the Appendix A.2. The pairwise sampling networks are used to generate samples of one random variable conditioned on a sample from a second variable. Within the inference algorithm, these conditional samples (line 6), are used to weight the samples of one variable using the unary information of a second variable (line 7).
>
> > The pull strategy approximation is adopted from Desingh et al. (2019). The quality of this approximation is not evaluated.
>
> > Further empirical approximations of the BP approach (which is itself an approximation) are explained in the supplement even though they are part of the proposed approach. Again, their approximation quality is not evaluated.
>
> The goal of this work was not to evaluate the approximation quality of the pull strategy adopted from Desingh et al. (2019), but rather to identify and validate a learning strategy to replace the hand-crafted factors required by this existing nonparametric belief propagation algorithm. To remain tractable in continuous state spaces, DNBP does use approximations. While the precise degree of approximation is not calculated directly, the comparisons to pure neural network baselines are intended to evaluate DNBP’s relevant performance characteristics in challenging problem domains.
>
> > The authors mention the work Xiong & Ruozzi, (2020), where the task of computing unary and pairwise marginals is solved approximately by variational inference and then combined with learning of MRF potentials given by neural networks. This approach has the potential to learn MRFs on graphs with cycles. I would have expected at least an experimental comparison with this method.
>
> We appreciate this suggestion to compare with the model proposed by Xiong & Ruozzi (2020). While this approach does learn the probabilistic factors for approximate MRF inference, it has only been demonstrated on discrete classification tasks and a two-mixture trivariate Gaussian dataset. In contrast, our work focuses on inference in continuous state spaces with a larger number of degrees of freedom. We agree that an interesting direction for future work would modify both approaches to allow for a direct comparison on high dimensional continuous state space inference tasks.
>
> > Furthermore, it is well known that supervised learning of MRFs with finite state spaces can be approached by using the Pseudo-likelihood estimator instead of the ML estimator. This obviates the need of computing unary and pairwise marginals during learning. I would expect that it is much easier to adapt the PL estimator to the case of infinite state spaces.
>
> We appreciate this suggestion to consider a pseudo-likelihood estimator. To the best of our knowledge, we are not aware of an existing method to use pseudo-likelihood estimation for belief propagation in continuous state spaces. We agree this could be an interesting direction for future work. We would greatly appreciate it if the reviewer can point us to related work that we could use.
>
> > The artificial examples studied in the experiments and seemingly also the human hand tracking use MRFs on trees. This would obviate the need for loopy BP, because in this case a standard two-pass BP can be used instead.
>
> We acknowledge that the graphs chosen for our experiments are trees, however it is possible that using loopy versions of the graphs could potentially improve DNBP’s performance by allowing the model to leverage additional pairwise relationships during inference. While alternative inference algorithms can be used on tree-based MRFs, such as two-pass BP, the inference process of DNBP is not specific to tree-based graphs.

---

> > ### Author Response · Authors · 2021-11-23
> > **Response to Reviewer Rbn6 (2/2)**
> >
> > > It seems that the proposed approach is worse (w.r.t. estimation) than the neural network baseline (Fig. 8)? This holds both, for the frame by frame variant and the tracking variant of the proposed approach? It remains unclear to me how the latter tracking variant works precisely.
> >
> > We acknowledge that the empirical performance of DNBP did not outperform the pure neural network baseline for hand tracking (Fig. 8). We have revised Section 5.6 to elaborate on the difference between the frame-by-frame and tracking variants of DNBP. The frame-by-frame variant performs iterative inference on each single frame of the dataset, ignoring the temporal relation between frames. In contrast, the tracking variant of DNBP performs iterative inference across frames of the dataset, allowing it to maintain belief over time.

---

> > ### Comment · Reviewer_Rbn6 · 2021-12-01
> > **Thank you**
> >
> > Thank you for the response and clarification of some of my questions and issues. Unfortunately, the main discrepancy I see in the paper remains unresolved: Is it a conceptual paper proposing a new differentiable  approach for supervised learning and prediction in MRFs with infinite state spaces? Then it would require a careful evaluation of all used approximations, a clear outline of its applicability area and a conceptual comparison with existing methods. Or is it a rather a novel empirical approach for solving a class of application problems? This would require thorough experiments, showing that the novel approach is better than existing methods.

---

### Official Review · Reviewer_yWQ2 · 2021-11-03

**Correctness:** 3
**Technical Novelty And Significance:** 3
**Empirical Novelty And Significance:** 2
**Recommendation:** 3
**Confidence:** 2

**Main Review:**

## Strengths

  - [S1] The proposed method produces meaningful uncertainty estimates,
    improving state estimation robustness under
    uncertainty/occlusion.
  - [S2] The writing is good.
  - [S3] The paper potentially opens up future avenues of
    work that build on the proposed framework, improving
    learning efficiency, tackling a wider range of
    problems, etc.


## Limitations

  - [L1] Limited empirical performance - the proposed
    approach seems to struggle to outperform a much simpler
    LSTM baseline. While the proposed DNBP does provide
    uncertainty estimates which are unavailable with the
    LSTM, there are many ways to add, train, and calibrate
    such outputs for NNs as well. However, such baselines
    are not evaluated.
  - [L2] The reasons why Hernando et al. outperform the
    proposed approach are not discussed at all, in spite of
    the large gap between the two methods.
  - [L3] Training is not end-to-end, as backpropagation
    does not occur through the particle resampling stage.
  - [L4] The comparisons to related papers which may, for
    example, also use NN factors but perform inference in a
    different way are a bit too brief in my opinion. For
    readers who are not experts in graphical models it may
    be difficult to establish what this paper does
    differently, and what its strengths and limitations are
    w.r.t. the rest of the literature.


## Suggestions (Per-Section)

### Related Work

  - Could you please elaborate on the differences between
    the setting of the current paper and that of, e.g.,
    Xiong & Ruozzi? That paper also seems to learn the deep
    potentials end to end. I did not read that paper in
    detail, but is the main idea that they use a
    variational method rather than belief propagation?


### Results

  - Can you please make the text on Fig. 3 a bit bigger,
    especially the legend? Currently the legend is
    difficult to read.



**Summary Of The Paper:**

  The paper proposes an extension of particle belief
  propagation which allows the factor parameters (e.g.,
  neural network weights for factors modeled as NNs) to be
  learned using standard stochastic gradient descent.

  This algorithm is then applied to several
  continuous-domain state estimation tasks involving
  articulated objects (e.g., hand pose estimation from
  images).

  The main contribution of the paper is adapting an
  existing line of nonparametric belief propagation methods
  to this setting, which allows (partial) end-to-end
  learning.

  The learning is only partly end-to-end as the particle
  resampling stage is non-differentiable. Instead, the
  proposed approach supervises the belief at every step of
  the algorithm, but does not backpropagate through the
  entire inference procedure opting to instead maximize the
  belief of the GT values for the unobserved labels at each
  step.

  Experimental results on synthetic and real datasets for
  articulated object state estimation show that the
  proposed algorithm is able to learn meaningful NNs for
  the unary and pairwise potentials while also producing
  reasonable uncertainty estimates. The proposed approach
  sometimes outperforms the baselines (e.g., an LSTM), but
  in the task of parametric human hand tracking, it lags
  behind the state of the art by a large margin.



**Summary Of The Review:**

  While I think that there is definitely a lot of potential
  in the paper, there are several areas that remain open to
  improvement. First, the comparisons to related work are a
  bit unclear to me, and explaining the trade-offs between
  different kinds of inference methods, especially through
  the lens of data-driven factor learning, would help
  position the paper better in this area of ML.

  Second, the empirical evaluation is not very thorough,
  and its results don't improve much over simple
  baselines like an LSTM. A broader evaluation, including
  on tasks where formulating "hand-engineered" potentials
  is more difficult would strengthen the paper in my
  opinion.

  Because of this I would recommend rejection for now
  unless the above issues are fully clarified in upcoming
  communications. That being said, I am likely to be the
  least experienced of this paper's reviewers when it comes
  to graphical models, so please take this into
  consideration when evaluating my reviews (this is
  addressed to both the authors, as well as to the ACs).

---

> ### Author Response · Authors · 2021-11-23
> **Response to Reviewer yWQ2**
>
> We thank Reviewer yWQ2 for their thorough feedback and address each point below.
> > Limited empirical performance - the proposed approach seems to struggle to outperform a much simpler LSTM baseline. While the proposed DNBP does provide uncertainty estimates which are unavailable with the LSTM, there are many ways to add, train, and calibrate such outputs for NNs as well. However, such baselines are not evaluated.
>
> We acknowledge that the empirical performance of DNBP did not outperform pure neural network baselines on the experiments performed. The goal of this work was not to demonstrate superior performance by DNBP over neural network baselines, but rather to identify and validate a learning strategy to replace the hand-crafted factors required by existing nonparametric belief propagation algorithms. We agree that based on the demonstrated results, future work should compare the estimates of uncertainty produced by DNBP to a neural network-based model of uncertainty.
>
> We strongly agree with your insight that the current results open the possibility for future lines of research to investigate how DNBP’s performance can be improved. However, in this work we focused on establishing the performance of an initial learning strategy rather than focusing on fine tuning to the particular datasets.
>
> > The reasons why Hernando et al. outperform the proposed approach are not discussed at all, in spite of the large gap between the two methods.
>
> Focusing on the performance of DNBP, we found that it predicted the pose of hands with less than 20mm of error on 45% of the test set. We found limited implementation details of the baseline from Garcia-Hernando et al. (2018), which is the reason why we did not propose specific explanations for DNBP’s relative performance difference.
>
> > Training is not end-to-end, as backpropagation does not occur through the particle resampling stage.
>
> We appreciate the observation that DNBP’s training approach is not end-to-end differentiable, across adjacent timesteps. As discussed in Section 4, this is due to the non-differentiable resampling operation used during the inference process. As discussed in our conclusion, this aspect of DNBP is left as a direction for future work.
>
> > The comparisons to related papers which may, for example, also use NN factors but perform inference in a different way are a bit too brief in my opinion. For readers who are not experts in graphical models it may be difficult to establish what this paper does differently, and what its strengths and limitations are w.r.t. the rest of the literature.
>
> We have revised the related work section to address this concern, in particular by expanding on the discussion of how DNBP compares to the approach proposed by Xiong & Ruozzi (2020). While this approach (Xiong & Ruozzi, 2020) is able to perform MRF inference using learned probabilistic factors, it has not been demonstrated on inference tasks in continuous state spaces which involve high dimensional random variables (i.e. random variables with more than 3 dimensions total). In contrast, our work focuses on inference in continuous state spaces with a larger number of degrees of freedom.

---

> > ### Comment · Reviewer_yWQ2 · 2021-11-24
> > **Response**
> >
> > Thank you for addressing the comments I posted.

---

### Decision · Program_Chairs · 2022-01-20

**Decision:**

Reject

**Comment:**

All reviewers concur that the paper has promise, but fails to deliver on that promise.  The idea of learning potentials based on DNNs is appreciated, but the evaluation of the contribution is considered lacking by all reviewers.  In addition, reviewers note that the training is not differentiable, which the rebuttal acknowledges is future work.

I do not reject the paper simply for failing to beat a deep learning baseline, but for having chosen applications which do not even test the paper's hypotheses: reviewers note that the models are tree structured, so loopy BP is not tested, despite the revised paper's claim that "the inference strategy is compatible with graphs containing cycles".